# Compressible-composable NeRF via Rank-residual Decomposition

**Jiaxiang Tang**[1], **Xiaokang Chen**[1], **Jingbo Wang**[2], **Gang Zeng**[1,3]
[1]School of Intelligence Science and Technology, Peking University
[2]Chinese University of Hong Kong   [3]Intelligent Terminal Key Laboratory of SiChuan Province
{tjx, pkucxk}@pku.edu.cn, wj020@ie.cuhk.edu.hk, zeng@pku.edu.cn

## Abstract

Neural Radiance Field (NeRF) has emerged as a compelling method to represent 3D objects and scenes for photo-realistic rendering. However, its implicit representation causes difficulty in manipulating the models like the explicit mesh representation. Several recent advances in NeRF manipulation are usually restricted by a shared renderer network, or suffer from large model size. To circumvent the hurdle, in this paper, we present a neural field representation that enables efficient and convenient manipulation of models. To achieve this goal, we learn a hybrid tensor rank decomposition of the scene without neural networks. Motivated by the low-rank approximation property of the SVD algorithm, we propose a rank-residual learning strategy to encourage the preservation of primary information in lower ranks. The model size can then be dynamically adjusted by rank truncation to control the levels of detail, achieving near-optimal compression without extra optimization. Furthermore, different models can be arbitrarily transformed and composed into one scene by concatenating along the rank dimension. The growth of storage cost can also be mitigated by compressing the unimportant objects in the composed scene. We demonstrate that our method is able to achieve comparable rendering quality to state-of-the-art methods, while enabling extra capability of compression and composition. Code is available at https://github.com/ashawkey/CCNeRF.

## 1 Introduction

Photo-realistic rendering and manipulation of 3D scenes have been long standing problems with numerous real-world applications, such as VR/AR, computer games, and video creation. Recently, the volumetric Neural Radiance Field (NeRF) representations [26, 1, 8, 27] show impressive progress in rendering photo-realistic images with rich details. However, due to this implicit representation of geometry and appearance, manipulating the underlying scenes encoded by NeRF still remains a challenging problem. To solve this problem, some works [21, 45, 19] introduce scene-specific features and scene agnostic rendering network, so that scenes trained with a shared rendering network can be composed together. However, the constrained and biased capability of these rendering networks causes difficulty in extending to various objects or scenes. New objects have to be trained with a fixed rendering network to be compatible with the old objects. Other works [36] discard the rendering network and adopt an no-neural-network NeRF representation, which is more convenient to manipulate the reconstructed scenes and is still able to render high-quality images. Nevertheless, the large storage requirement for each single model is detrimental to composing complex scenes with lots of objects.

We present a novel approach that allows efficient and convenient manipulation of scenes represented with our model. Two aspects should be fulfilled to achieve this goal. The first is that we can dynamically adjust the model size to support different levels of detail (LOD) in different scenarios.

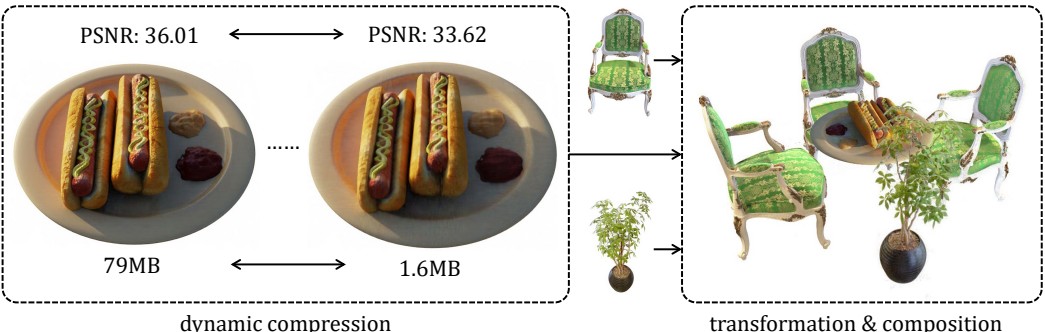

PSNR: 36.01 ←→ PSNR: 33.62

79MB ←→ 1.6MB

dynamic compression

transformation & composition

Figure 1: **Compressibility and Composability of our method.** We present a tensor rank decomposition based neural field representation, which supports model compression through rank truncation, and arbitrary composition between different models through rank concatenation. Both of these operations require **no extra optimization**, or any constraints in training (*e.g.*, a shared renderer).

This functions similar to mipmaps in graphics and requires no extra optimization step. The second is that all models can be transformed and composed arbitrarily for manipulation with no constraints in training. This promises that our models are always reusable, and support the most basic operations in a 3D editor like blender [9]. We name these two properties as compressibility and composability.

For the **compressibility**, we are motivated by the properties of Singular Value Decomposition (SVD) and High-order SVD (HOSVD) [10]. Our aim is to learn the decomposition of a 3D scene from only 2D observations like TensoRF [8], and further preserve the near-optimal low-rank approximation property. We propose a simple and flexible tensor rank decomposition based neural radiance field, and a rank-residual learning strategy. Each 3D scene is modeled by a 4D feature volume, which can be described with a set of rank components and a matrix storing the weights for each feature channel. The rank components are either vector- or matrix-based, corresponding to the CANDECOMP/PARAFAC (CP) decomposition [14, 4] and a less compact triple plane variant. We introduce a rank-residual learning strategy to encourage the lower ranks to preserve more important information of the whole scene. Combined with an empirical sort-and-truncate strategy, the proposed method achieves near-optimal low-rank approximation at any targeted rank. Different LODs are represented with different low-rank truncations of the model, allowing dynamic trade-off between model size and rendering quality without retraining. Besides, our model contains no neural networks and thus naturally supports **composability**. Since there are no MLP renderers in our model, we can compose different objects by simply concatenating their rank components. A transformation matrix is recorded for each object to control its position and orientation in the scene.

As demonstrated in Figure 1, we are able to control each model's LOD and size in a flexible range, and perform arbitrary transformation and composition of different models. Furthermore, these two properties are connected together through the underlying concept of rank, and can be combined in practical use. For example, we can mitigate the growth of model size of a complex scene composed of multiple objects, by compressing the less important objects. Our contributions can be summarized as follows: (1) We propose a simple radiance field representation based on two types of tensor rank decomposition, which allows flexible control of model size and naturally supports transformation and composition of different models. (2) We design a rank-residual learning strategy to enable near-optimal low-rank approximation. After training, our model can be dynamically adjusted to trade off between performance and model size without retraining. (3) The proposed method reaches comparable rendering quality with state-of-the-arts, while additionally enabling both compressibility and composability.

## 2 Related Work

### 2.1 Scene Representation with NeRF

3D scenes can be represented with various forms, including volumes, point clouds, meshes, and implicit representations [33, 34, 7, 30, 38, 25, 23, 28]. NeRF [26] proposes to use a 5D function

to represent the scene and applies volumetric rendering for novel view synthesis, achieving photo-realistic results and detailed geometry reconstruction. This powerful representation quickly receives attention and is extensively studied and applied in various fields [50, 24], such as generative settings [6, 37, 29, 5], dynamic scenes [20, 31], and texture mapping [44]. In particular, we categorize recent progress by the design of the underlying functions into three classes: neural network-based, hybrid and no-neural-network. neural network-based representations typically apply an MLP, as the implicit function to encode 3D scenes. The original NeRF [26] and most following works [1, 2, 51, 43, 46, 35] choose this representation for its simplicity. However, the training and inference speed of such a network is generally slow due to the relatively expensive MLP computation. Therefore, hybrid representations try to reduce the size of the MLP, by storing the 3D features in an explicit data structure. Since a dense 3D representation is unaffordable, different methods are explored. For example, NSVF [21] adopts sparse voxel grids, PlenOctrees [49] adopts octrees, instant-ngp [27] adopts a multi-scale hashmap, and TensoRF [8] factorizes the scene into lower-rank components. Querying such hybrid representation is much faster, thus reducing training and inference time and even reaching interactive FPS. Lastly, no-neural-network representations attempt to model the 3D scene without neural networks. Plenoxels [36] shows that only the explicit sparse voxels representation is enough to model complex 3D scenes. Our method also belongs to this representation, sharing the similar tensor rank decomposition idea to TensoRF [8], but we focus on two additional capabilities, *i.e.*, compressibility and composability, which are important yet usually absent in previous work.

## 2.2  Tensor Decomposition and Low-rank Approximation

Decomposition of high-order tensors [18] can be considered as the generalizations of matrix singular value decomposition. The Tucker decomposition [40] decomposes a tensor into a core tensor multiplied by a matrix along each mode. The CANDECOMP/PARAFAC (CP) decomposition [14, 4] factorizes a tensor into a sum of component rank-one tensors, and can be viewed as a special case of Tucker where the core tensor is superdiagonal. The high-order singular value decomposition [10] provides a method to compute a specific Tucker decomposition with an all-orthogonal core tensor. Low-rank approximation is a common problem that applies tensor decomposition, and has found various applications such as image compression. Although the truncated HOSVD does not hold the optimal property contrary to the truncated SVD, it still results in a quasi-optimal solution [10, 41, 12], which is enough to yield a sufficiently good solution in practical uses. Tensor rank decomposition and its variants [10, 11] has been used in various vision and learning tasks [47, 8, 48]. Specifically, TensoRF [8] first leverages the CP decomposition and a Vertex-Matrix (VM) decomposition to factorize neural radiance fields, but its other designs (*e.g.*, use of MLP) disturbs the property of tensor rank decomposition and prevents it from achieving compression or composition. Instead, we focus on modeling neural radiance fields only with tensor rank decomposition, and aim to preserve the low-rank approximation property, enabling the compression of a learned neural radiance field similar to the SVD compression of an image.

## 2.3  Manipulation and Composition of NeRF

Manipulation and Composition are important for a 3D representation's practical usage. Explicit 3D representations, *e.g.*, meshes, are natively editable and composable. However, neural network-based implicit representations like a vanilla NeRF is difficult to perform such operations. NSVF [21] can composite separate objects together, but these objects have to be trained together using a shared MLP, which limits its flexibility and potential usage. Later works [45, 29, 13, 19] learn object-compositional NeRF, but are usually scene-specific and do not allow cross-scene composition without retraining. Geometry and appearance editing [22, 42] of neural fields also requires an extra optimization step to modify the neural network-based representation. With the explicit sparse voxel representation, Plenoxels [36] naturally supports direct composition of different objects, but suffers from the large storage on the dense index matrix. Our method also supports arbitrary affine transformations and compositions without extra optimization. Further, we can efficiently mitigate the model size growth due to the compact tensor rank decomposition and the compressibility.

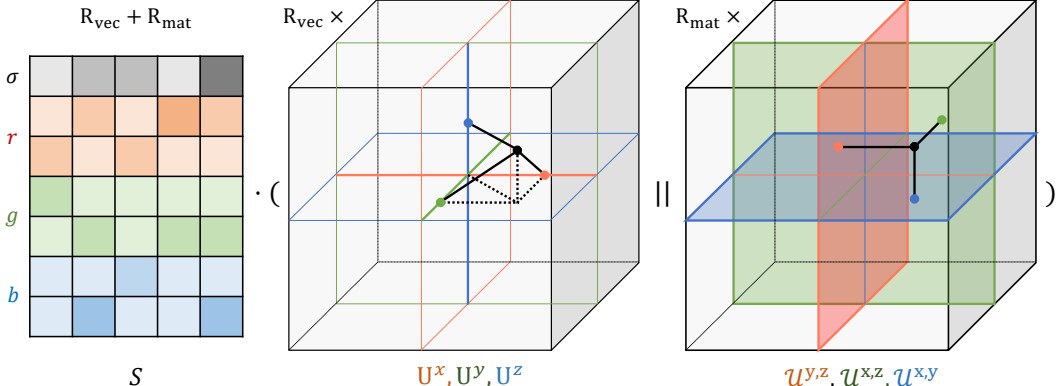

Figure 2: **Model structure.** Our model is composed of a matrix storing rank weights for different feature channels, and a set of decomposed rank components. Each rank component can be either vector- or matrix-based, and the ratio can be controlled to trade off between model size and performance. To query any 3D coordinate, we first project it to the decomposed vectors or matrices as denoted by the black lines, and then perform weighted interpolation. || denotes concatenation along the rank dimension.

## 3 Methodology

### 3.1 Preliminaries on Neural Radiance Fields

Neural Radiance Fields (NeRF) [26] represents a 3D volumetric scene with a 5D function $f_\Theta$ that maps a 3D coordinate $\mathbf{x} = (x, y, z)$ and a 2D viewing direction $\mathbf{d} = (\theta, \phi)$ into a volume density $\sigma$ and an emitted color $\mathbf{c} = (r, g, b)$. Given a ray $\mathbf{r}$ originating at $\mathbf{o}$ with direction $\mathbf{d}$, we query $f_\Theta$ at points $\mathbf{x}_i = \mathbf{o} + t_i \mathbf{d}$ sequentially sampled along the ray to get densities $\{\sigma_i\}$ and colors $\{\mathbf{c}_i\}$. The color of the pixel corresponding to the ray is then estimated by numerical quadrature:

$$\hat{\mathbf{C}}(\mathbf{r}) = \sum_i T_i \alpha_i \mathbf{c}_i, T_i = \prod_{j<i}(1 - \alpha_j), \alpha_i = 1 - \exp(-\sigma_i \delta_i), \delta_i = t_{i+1} - t_i \tag{1}$$

where $\delta_i$ is the step size, $\alpha_i$ is the opacity, and $T_i$ is the transmittance. Since this volume rendering process is differentiable, NeRF can be optimized only from 2D image supervision by minimizing the L2 difference between each pixel's predicted color $\hat{\mathbf{C}}(\mathbf{r})$ and the ground truth color $\mathbf{C}(\mathbf{r})$ from the image:

$$\mathcal{L}_{\text{NeRF}} = \sum_{\mathbf{r}} ||\mathbf{C}(\mathbf{r}) - \hat{\mathbf{C}}(\mathbf{r})||_2^2 \tag{2}$$

### 3.2 Preliminaries on Tensor Decomposition

For a 3D tensor $\mathcal{T} \in \mathbb{R}^{H \times W \times D}$, each element $\mathcal{T}_{i,j,k} \in \mathbb{R}$ can be represented via the Tucker decomposition [40] by:

$$\mathcal{T}_{i,j,k} = \sum_{p=1}^{P} \sum_{q=1}^{Q} \sum_{r=1}^{R} \mathcal{S}_{p,q,r} \mathbf{U}_{i,p}^x \mathbf{U}_{j,q}^y \mathbf{U}_{k,r}^z \tag{3}$$

where $\mathcal{S} \in \mathbb{R}^{P \times Q \times R}$ is the core tensor, $P, Q, R$ are the number of components along each axis, and $\mathbf{U}^x \in \mathbb{R}^{H \times P}, \mathbf{U}^y \in \mathbb{R}^{W \times Q}, \mathbf{U}^z \in \mathbb{R}^{D \times R}$ are the factor matrices. The CP decomposition can be viewed as a special case of Tucker when $P = Q = R$ and $\mathcal{S}$ is superdiagonal [18] (*i.e.*, $\mathcal{S}_{i,j,k} \neq 0 \iff i = j = k$):

$$\mathcal{T}_{i,j,k} = \sum_{r=1}^{R} \mathbf{s}_r \mathbf{U}_{i,r}^x \mathbf{U}_{j,r}^y \mathbf{U}_{k,r}^z \tag{4}$$

where $\mathbf{s} = \text{diag}(\mathcal{S}) \in \mathbb{R}^R$ is the reduced core tensor (or rank weights). Although $\mathbf{s}$ is usually absorbed into the factor matrices, we write it out for the convenience of later discussion.

## 3.3 Decompose NeRF without MLP

**Hybrid Feature Volume Decomposition.** We are interested in multi-feature volumetric encoded as a 4D tensor $\mathcal{T} \in \mathbb{R}^{C \times H \times W \times D}$, where $C$ is the feature dimension (*e.g.*, density, RGB values, or other features), and $(H, W, D)$ is the spatial resolution (usually $C \ll \min(H, W, D)$). A straightforward way is to perform $C$ independent decompositions for each channel. However, different feature channels such as the RGB values are highly correlated in real 3D scenes. A more compact way is to share the factorized matrices like TensoRF [8], and only use different rank weights $\mathbf{s}$ for different channels. Therefore, we propose to represent $\mathcal{T}$ through the CP decomposition by:

$$\mathcal{T}_{i,j,k} = \mathbf{S} \cdot (\mathbf{U}_i^x * \mathbf{U}_j^y * \mathbf{U}_k^z) \tag{5}$$

where $\mathbf{S} \in \mathbb{R}^{C \times R}$ is the matrix of rank weights for $C$ channels, and $*$ denotes the Hadamard product. Since the above decomposition relies on rank-one vectors (1D tensors), it may require very high ranks to represent complex 3D scenes, which leads to expensive computation at each location. A less compact but more computation-friendly alternative is to adopt matrices (2D tensors) to factorize the 3D scene. This variant in the form of CP decomposition is given by:

$$\mathcal{T}_{i,j,k} = \mathbf{S} \cdot (\mathcal{U}_{i,j}^{x,y} * \mathcal{U}_{j,k}^{y,z} * \mathcal{U}_{i,k}^{x,z}) \tag{6}$$

where $\mathcal{U}^{x,y} \in \mathbb{R}^{H \times W \times R}, \mathcal{U}^{y,z} \in \mathbb{R}^{W \times D \times R}, \mathcal{U}^{x,z} \in \mathbb{R}^{H \times D \times R}$ are the factorized matrices along three planes, each containing $R$ components. This variant can be comprehended by first slicing and tiling the original 3D space along each axis, and then learning a CP decomposition on $\mathbb{R}^{HW \times WD \times HD}$. Although this representation is less compact and takes more storage, recent works [8, 5] have shown that it is able to represent scenes with smaller $R$ and better quality. We denote this variant as the Triple Plane (TP) decomposition. Further, we notice that for each individual rank, the underlying vector- or matrix-based decomposition can be selected independently. Therefore, a hybrid variant (HY) that combines the above CP and TP decomposition is proposed. We can flexibly adjust the ratio of two decompositions by $R = R_{\text{vec}} + R_{\text{mat}}$ to trade off between model size and performance. The model structure is illustrated in Figure 2.

**Learning the Decomposition via Differentiable Rendering.** For simplicity, we take the CP decomposition as an example. Our model only consists of four tensors to optimize, *i.e.*, $\mathbf{S}, \mathbf{U}^x, \mathbf{U}^y, \mathbf{U}^z$. To represent scenes through neural radiance fields, we need to learn the volume density $\sigma$ and color $\mathbf{c}$ at each location. As the volume density is only dependent on the 3D coordinate $\mathbf{x}$, we can represent it with one feature channel $\mathcal{T}_{i,j,k}^{\text{density}} \in \mathbb{R}$. However, the color is dependent on both the 3D location $\mathbf{x}$ and the 2D viewing direction $\mathbf{d}$, which is a 5D function in total. To represent it within the 3D feature volumes, we adopt the spherical harmonics (SH) functions to approximate the additional 2D viewing directions dependency [49, 36]. In particular, for spherical harmonics of maximum degree $\ell_{\max}$, it takes $(\ell_{\max} + 1)^2$ SH coefficients to model the view-dependent color per channel. We use $\mathcal{T}_{i,j,k}^{\kappa} \in \mathbb{R}^{(\ell_{\max}+1)^2}, \kappa \in \{r, g, b\}$ to represent these coefficients. The density and color at each location (coordinate indices are omitted for simplicity) can then be represented with:

$$\sigma = \phi(\mathcal{T}^{\text{density}}); c^{\kappa} = \psi(\sum_{\ell=0}^{\ell_{\max}} \sum_{m=-\ell}^{\ell} \mathcal{T}_{\ell,m}^{\kappa} Y_{\ell}^m(\mathbf{d})), \kappa \in \{r, g, b\} \tag{7}$$

where $\phi(\cdot)$ is the density activation, $\psi(\cdot)$ is the color activation, $\mathbf{d}$ is the viewing direction, and $Y_{\ell}^m$ are the SH functions. Our model can then be optimized through the standard RGB loss in Equation 2.

## 3.4 Rank-residual Learning for Compressibility

The idea of low-rank approximation is to only keep the most important rank components, where the importance of each rank component can be represented by the singular values in the SVD algorithm. Similarly, we can define the importance of each rank component in our decomposition by the rank weights $\mathbf{S}$ averaged on all feature channels, and multiplied with the magnitude of three factor matrices along the rank dimension. However, the CP decomposition doesn't hold this low-rank approximation property [18]. If we directly sort the rank components by the rank importance and truncate the model for compression, the rendering quality drops sharply compared to the optimal model (*i.e.*, model retrained with the same parameters), as illustrated by the baseline method in Figure 3.

We propose a rank-residual learning strategy to close this performance gap and achieve near-optimal compression results. This strategy aims to simulate SVD's low-rank approximation property, where

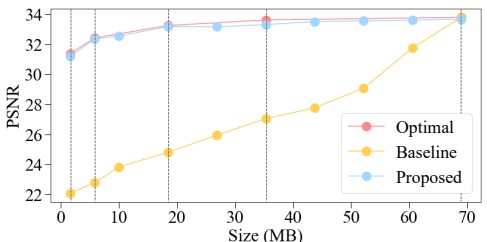

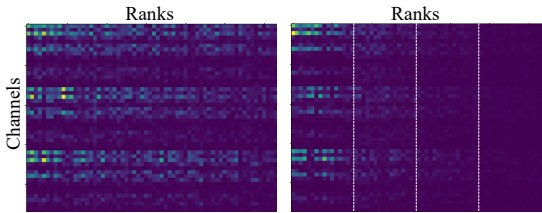

Figure 3: **Compression at any rank.** Combined with the empirical sort-and-truncate strategy, the proposed model achieves near-optimal compression at any rank. We use the HY-S model on the LEGO dataset as an example, and the dashed lines indicate where we apply rank-residual supervision.

Figure 4: **Visualization of rank importance.** Ranks are sorted column-wisely based on the averaged rank importance. The rank importance is more concentrated in the proposed method (right) compared to the baseline (left), which is crucial for truncation-based compression. We use the HY model on the LEGO dataset as an example, and the dashed lines indicate where we apply rank-residual supervision.

the lower rank components contain more information and contribute more to the approximation. Since the rank components in our method can be flexibly adjusted, a direct solution is to train in a progressive way. Suppose the total number of ranks is $R$, and the number of training stages is $M$. The total $R$ rank components can be sequentially divided into $M$ non-empty groups, and we denote the accumulated number of ranks for each group by $R_m$ where $m \in \{1, 2, \cdots, M\}$. We start from training the first $R_1$ rank components, and after its convergence, we fix them and append the next $R_2 - R_1$ rank components to train in a new stage. In the last stage, all $R$ ranks are involved. However, this is inefficient since we need to make sure each stage is fully converged before increasing the number of ranks. A more efficient way is to train all stages in parallel. In particular, we simultaneously supervise the sequentially accumulated outputs from all $M$ groups. This can be viewed as supervising the outputs from $M$ truncations of the decomposition with a rank-residual loss:

$$\mathcal{L}_{\text{residual}} = \sum_{\mathbf{r} \in \mathcal{R}} \sum_{m=1}^{M} ||\mathbf{C}(\mathbf{r}) - \hat{\mathbf{C}}_m(\mathbf{r})||_2^2 \tag{8}$$

where $\hat{\mathbf{C}}_m(\mathbf{r})$ is RGB values calculated from the truncated decomposition $\mathbf{S}_m, \mathbf{U}_m^x, \mathbf{U}_m^y, \mathbf{U}_m^z$ which only keeps the first $R_m$ rank components (taking the CP decomposition as an example). During training, each group is able to learn the residual error from previous groups, and eventually leads to the desired low-rank approximation property. Ideally, choosing $M = R$ groups assures this low-rank approximation property to hold at any targeted rank, but usually computationally unaffordable with a large $R$. In practice we choose $M \ll R$, so the property only holds at those dividing ranks $\{R_m\}$. For any other rank $R'$, assuming $R_m < R' < R_{m+1}$, we first keep the fully covered $R_m$ rank components, and then use the empirical sort-and-truncate strategy to select the remaining $R' - R_m$ ranks in the last group $\{R_m + 1, R_m + 2 \cdots, R_{m+1}\}$. By selecting the top rank components with the largest average importance, it is enough to keep the near-optimal low-rank approximation property at any targeted ranks, as shown by the proposed method in Figure 3.

The decomposition model trained with our rank-residual learning allows dynamic adjustment of model size and rendering quality with no extra optimization. In practice, we usually need different LODs to adapt to different cases, such as the texture mipmaps. While the other NeRF representations need to retrain for different LODs and store every model separately, we just train once and get a unified model for all LODs. Given a targeted storage upper bound or performance lower bound, we only need to select the targeted rank and truncate the decomposition with the simple slicing operation.

### 3.5 Composability without Constraints

As illustrated in Figure 5, any 3D object or scene represented by our model can be composed without the constraints in previous work [21, 45]. Since our model describes each 3D scene with a set of rank components, composability is naturally accomplished by concatenating along the rank dimension and summing up the number of ranks $R = \sum_{n=1}^{N} R_n$, where $n$ denotes the object index. In practice, this is implemented by appending new rank components to a parameter list, so that models with different

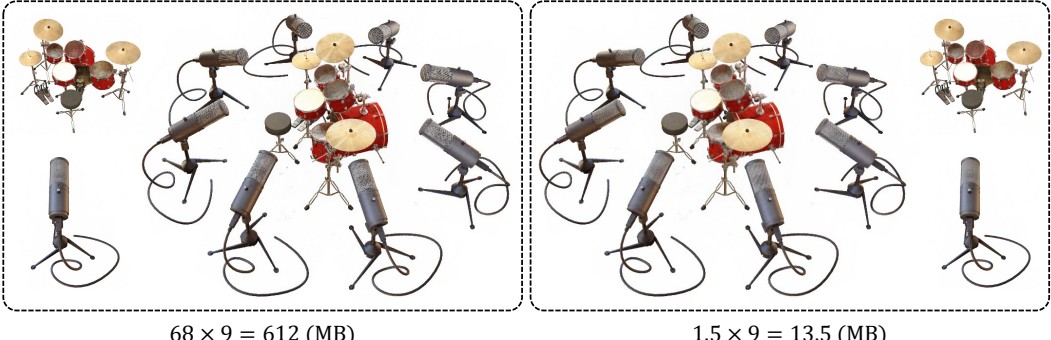

| $68 \times 9 = 612$ (MB) | $1.5 \times 9 = 13.5$ (MB) |

Figure 5: **Compressing a scene composed of multiple objects.** For a scene composed of lots of different objects, we can compress the less important objects to achieve better efficiency and less storage with a little sacrifice of rendering quality.

resolution and decomposition forms can be composed together. For each object, we record its number of ranks so we can still distinguish it from the whole scene. Arbitrary affine transformations of individual objects are supported by recording a transformation matrix $\mathbf{T}_n \in \mathbb{R}^{4 \times 4}$ for each object, which includes the translation $\mathbf{t}_n \in \mathbb{R}^3$, rotation $\mathbf{R}_n \in \mathrm{SO}(3)$ and scale $\mathbf{s}_n \in \mathbb{R}^3$. We warp the ray into each object's coordinate system before querying the density $\{\sigma_{i,n}\}$ and color $\{\mathbf{c}_{i,n}\}$ at the sampled points:

$$\sigma_{i,n}, \mathbf{c}_{i,n} = f_\Theta(\mathbf{T}_n \mathbf{x}_i, \mathbf{R}_n \mathbf{d}) \tag{9}$$

Here we assume the coordinate $\mathbf{x}_i$ is homogeneous, and the viewing direction $\mathbf{d}$ is represented by a unit vector. To correctly handle the occlusion between different objects, we still sample one ray per pixel, and perform the composition at each sample point by:

$$\begin{cases} \alpha_i = 1 - \exp(-\delta_i \sum_{n=1}^N \sigma_{i,n}) \\ \mathbf{c}_i = \sum_{n=1}^N \varphi_N(\sigma_{i,n}) \mathbf{c}_{i,n} \end{cases} \tag{10}$$

We sum up the density from all objects to calculate opacity, and weight the color by the density after the softmax function $\varphi_N$. The rendering formula in Equation 1 can then be applied to calculate the pixel color. Although the model size and rendering time grows linearly with the total number of ranks (complexity of the scene), we show in experiments that the compression property can be applied to mitigate the growth and improve efficiency.

## 4 Experiments

### 4.1 Implementation Details

The model is implemented with the `PyTorch` framework [32]. The degree for SH coefficients is 3, which equals 48 channels for the color feature. We use the Adam optimizer [16] with an initial learning rate of 0.02 for the factorized matrices, and 0.001 for the singular values. All the experiments are performed on one NVIDIA V100 GPU. The resolution of the feature grid is determined by the total number of voxels $N$ and the bounding box, where $N$ is increased from $128^3$ to $300^3$ for HY models and $500^3$ for CP models in early training steps. To accelerate rendering, we adopt the binary occupancy mask pruning technique as in [8, 27], and use separate rank components for density and color to avoid unnecessary querying in empty space. This occupancy mask is also used to shrink the initial bounding box for more precise modeling. We mainly carry out experiments on the NeRF-synthetic dataset [26] (CC BY 3.0 license) and the Tanks and Temples dataset [17] (CC BY-NC-SA 3.0 license). Please check the supplementary materials for more details.

### 4.2 Compression Results

Firstly, We evaluate the compressibility of our model. Since the color components cost most of the total storage, we mainly focus on compressing the color components, and keep the density components fixed. Given the number of ranks $R$, we first train two models with the original loss



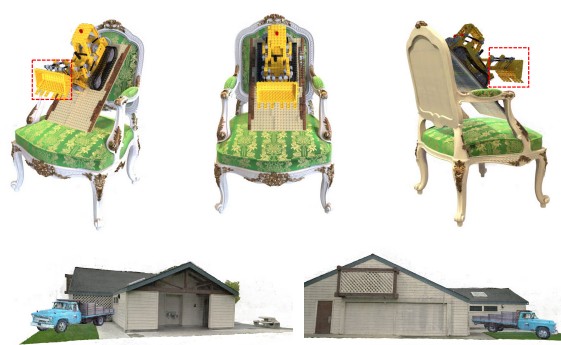

Figure 6: **Visualization of compression.** The baseline method deteriorates significantly, while our proposed method remains high rendering quality.

Figure 7: **Visualization of composition.** We show composition between different models with our method. Our method can successfully handle occlusion and remain high-quality rendering.

Table 1: **Compression Results.** We report the PSNR for different compression strategies. CP, HY and HY-S denotes three model settings with different range of ranks. Ranks are denoted by $R_{\text{vec}}^{\text{density}}/R_{\text{mat}}^{\text{density}}$-$R_{\text{vec}}^{\text{color}}/R_{\text{mat}}^{\text{color}}$. We emphasize how the proposed method improves over the truncate baseline, especially at highly compressed conditions.

| Model | Ranks | Resolution | Size (MB) | Optimal | Baseline | Proposed |
|-------|-------|------------|-----------|---------|----------|----------|
| CP | 96/0-384/0 | 500 | 4.4 | 30.78 | 30.78 | 30.55 (−0.23) |
| | 96/0-288/0 | 500 | 3.8 | 30.68 | 28.78 | 30.46 (+**1.68**) |
| | 96/0-192/0 | 500 | 3.2 | 30.38 | 26.95 | 30.15 (+**3.20**) |
| | 96/0-96/0 | 500 | 2.7 | 29.78 | 24.97 | 29.53 (+**4.56**) |
| HY-S | 96/0-96/64 | 300 | 68.9 | 31.54 | 31.54 | 31.22 (−0.32) |
| | 96/0-96/32 | 300 | 35.2 | 31.36 | 27.57 | 31.09 (+**3.52**) |
| | 96/0-96/16 | 300 | 18.4 | 31.04 | 25.66 | 30.83 (+**5.17**) |
| | 96/0-96/4 | 300 | 5.7 | 30.40 | 24.24 | 30.13 (+**5.89**) |
| | 96/0-96/0 | 300 | 1.5 | 29.49 | 23.54 | 29.30 (+**5.76**) |
| HY | 64/16-256/64 | 300 | 88.0 | 32.43 | 32.43 | 32.36 (−0.07) |
| | 64/16-192/48 | 300 | 70.8 | 32.42 | 30.63 | 32.35 (+**1.72**) |
| | 64/16-128/32 | 300 | 53.7 | 32.31 | 28.50 | 32.29 (+**4.09**) |
| | 64/16-64/16 | 300 | 36.5 | 31.96 | 26.30 | 31.94 (+**5.64**) |

$\mathcal{L}_{\text{NeRF}}$ in Equation 2 and our rank-residual loss $\mathcal{L}_{\text{residual}}$ in Equation 8. We denote them as $\mathcal{M}_R$ and $\mathcal{M}_R^{\text{ours}}$, respectively. At any targeted rank $r \leq R$ to compress, we design three strategies to verify whether the proposed compression is near-optimal: (1) Retrain a model $\mathcal{M}_r$ with $\mathcal{L}_{\text{NeRF}}$ at the given rank. This requires a retraining from scratch, and can be viewed as the optimal compression result at the given rank. (2) Sort and truncate the rank of the original model $\mathcal{M}_R$ to $\mathcal{M}_r^{\text{base}}$, which can be viewed as the baseline compression result. (3) Sort and truncate the rank of the proposed model $\mathcal{M}_R^{\text{ours}}$ to $\mathcal{M}_r^{\text{ours}}$. We denote these three settings as 'Optimal', 'Baseline', and 'Proposed' respectively.

The quantitative results are listed in Table 1. We find that the performance of the baseline method degrades significantly compared to the optimal method, whereas our proposed method is comparable to the optimal method at all targeted ranks. The visualization in Figure 6 demonstrates how the baseline model gradually deteriorates compared to the proposed model. Note that at the the right-most column where the baseline model is not compressed and the same as the optimal model, the rendering quality of the proposed model is hard to discern from the optimal model. In Figure 3, we show that even though our model is only supervised at 5 discrete ranks, it can remain good compression quality at any other ranks. Figure 3 provides an explanation for the compressibility of our model. With the rank-residual learning, the rank importance is more concentrated to the lower ranks, which benefits the low-rank approximation.

Table 2: **Comparison with recent methods.** Our method achieves comparable results while enabling both capability of compression and composition.

| | | Capability | | Synthetic-NeRF | | TanksTemples | |
|---|---|---|---|---|---|---|---|
| Method | Size (MB) | Composable | Compressible | PSNR↑ | SSIM↑ | PSNR↑ | SSIM↑ |
| SRN [38] | - | ✗ | ✗ | 22.26 | 0.846 | 24.10 | 0.847 |
| NeRF [26] | 5.0 | ✗ | ✗ | 31.01 | 0.947 | 25.78 | 0.864 |
| NSVF [21] | - | ✓ | ✗ | 31.75 | 0.953 | 28.48 | 0.901 |
| SNeRG [15] | 1771.5 | ✗ | ✗ | 30.38 | 0.950 | - | - |
| PlenOctrees [49] | 1976.3 | ✓ | ✗ | 31.71 | - | 27.99 | 0.917 |
| Plenoxels [36] | 778.1 | ✓ | ✗ | 31.71 | - | 27.43 | 0.906 |
| DVGO [39] | 612.1 | ✗ | ✗ | 31.95 | 0.975 | 28.41 | 0.911 |
| TensoRF-CP-384 [8] | 3.9 | ✗ | ✗ | 31.56 | 0.949 | 27.59 | 0.897 |
| TensoRF-VM-192 [8] | 71.8 | ✗ | ✗ | 33.14 | 0.963 | 28.56 | 0.920 |
| Instant-NGP [27] | 63.3 | ✗ | ✗ | 33.18 | - | - | - |
| Ours-CP | 4.4 | ✓ | ✓ | 30.55 | 0.935 | 27.01 | 0.878 |
| Ours-HY-S | 68.9 | ✓ | ✓ | 31.22 | 0.947 | 27.52 | 0.900 |
| Ours-HY | 88.0 | ✓ | ✓ | 32.37 | 0.955 | 28.08 | 0.913 |

## 4.3  Composition Results

In Figure 7, we demonstrate the composability of our method. Without extra optimization, we are able to perform affine transformation and composition of different models like composing meshes in a 3D editor. Note that our model can correctly handle the occlusion and collision between different objects. In Figure 5, we combine the compression capability of our model in scene composition. In a complex scene composed of lots of objects, we can compress the less important objects to make a trade-off between the model size and rendering quality.

## 4.4  Comparisons and Discussion

**Rendering Quality.** We compare our method with some recent works in Table 2. We focus on the extra capabilities to facilitate practical applications, but not boosting the rendering quality over the previous state-of-the-arts, since the vanilla NeRF [26] already reaches photo-realistic rendering in most cases. Although the performance of the proposed method is not the best, the simple model design with the extra compressibility and composability is unique and enables various applications.

**MLP Renderers.** As discussed in [8, 27], the absence of a small MLP renderer network generally leads to worse performance, especially for the specular details. However, these renderers are trained separately and cannot be shared across scenes unless explicitly restricted, which causes inconvenience or limits the potency of composition. We therefore discard the MLP at the cost of a slightly worse rendering quality, but facilitates the composition and compression.

**Limitations.** Although our method can correctly compose the geometry of multiple objects, we don't consider the lighting conditions. Our model bakes lighting conditions into the color like the vanilla NeRF, and cannot perform re-lighting to achieve consistent lighting effect after composition. A future direction is to integrate the reflectance models [3, 52]. Besides, we only support modeling bounded objects for now. A background model [51] can be combined to simulate unbounded scenes.

## 5  Conclusion

In this work, we present a novel compressible and composable neural radiance field representation. Our model is designed to be simple and flexible, yet still being effective enough to render photo-realistic images. A rank-residual learning strategy is proposed to enable near-optimal low-rank approximation, which allows dynamic adjustment of the model size to support different LODs in different scenarios. All models represented with our method can be arbitrarily transformed and composed together like in a 3D editor. Powered by these properties, we are able to efficiently and conveniently manipulate complex scenes with multiple objects. We believe our method will further facilitate the NeRF-based scene representation in real-world applications.

**Acknowledgements.** This work is supported by the National Key Research and Development Program of China (2020YFB1708002), National Natural Science Foundation of China (61632003, 61375022, 61403005), Grant SCITLAB-20017 of Intelligent Terminal Key Laboratory of SiChuan Province, Beijing Advanced Innovation Center for Intelligent Robots and Systems (2018IRS11), and PEK-SenseTime Joint Laboratory of Machine Vision.

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
