# OpenReview forum: "Compressible-composable NeRF via Rank-residual Decomposition"
_NeurIPS.cc/2022/Conference — NeurIPS 2022 Accept_

### Official Review · Reviewer_x5VX · 2022-07-08

**Rating:** 5
**Confidence:** 4
**Soundness:** 3 good
**Presentation:** 3 good
**Contribution:** 3 good

**Summary:**

This paper aims to compress neural field representations. To this end, the authors first propose a hybrid tensor decomposition and learn the decomposition via differentiable rendering. To encourage the primary information be learned in lowewr ranks, they introduce a novel training strategy that gradually increase the rank components to approximate the scene content.

**Questions:**

Please fix the problems in the weaknesses, which can improve the paper quality.

The property of this approach that impresses me is enabling the explicit controll of model capacity. Analysis on this property will make me feel better to this method.

**Ethics Review Area:**

["I don’t know"]

**Limitations:**

Yes.

**Strengths And Weaknesses:**

Strengths

- Overall, I feel the proposed approach is novel and sound. Compared with NeRF, this approach can explicitly add rank components to increase the model capacity, enabling them to dynamically adjust the model size during training.  This is a nice property.
- The paper is well-written and clearly presented.
- The comparison experiments and ablation studies are sufficient.

Weaknesses

1. Experiments

- Current experiments do not show that the proposed model has obvious advantages than NeRF or Mip-NeRF 360. What is the training time of the proposed approach.
- How to better analyze the property that you can gradually increase the rank components to improve the model capacity. An ablation study on this property will significanly bring readers more insights.

2. Writting

- Table 2 says that NeRF is not composable. I do not think so. As shown in [1, 2], object-centric NeRFs can be composed. A single NeRF model can even be explicitly manipulated, as shown in [3].

[1] Object-Centric Neural Scene Rendering
[2] Neural Scene Graphs for Dynamic Scenes
[3] NeRF-Editing: Geometry Editing of Neural Radiance Fields

---

> ### Author Response · Authors · 2022-08-02
> **Reply to reviewer x5VX**
>
> We thank the reviewer for the constructive suggestions. Below are our responses to the questions.
>
> **Q1: Training time of the proposed approach.**
>
> **A1**: The training of our CP, HY-S, and HY model costs 29, 30, and 41 minutes respectively, similar to TensoRF [1]. The convergence speed is considerably faster compared to the vanilla NeRF and Mip-NeRF 360, which take hours or days to converge. This speed advantage is mainly from the no-neural-network NeRF representation we use.
>
> **Q2: Advantages compared to NeRF or Mip-NeRF 360.**
>
> **A2**: (1) As shown in Table 2, our HY-S and HY model achieve better PSNR compared to NeRF on both datasets, and our CP model achieves smaller model size compared to NeRF. (2) Our model supports dynamic compression and natural composition, while the vanilla NeRF and Mip-NeRF 360 cannot be adjusted or composed after training. (3) As discussed in Q1, our model also achieves much faster convergence speed compared to the vanilla NeRF and Mip-NeRF 360.
>
> **Q3: Object-centric NeRFs can also be composed, and Edit-NeRF can manipulate a single NeRF model.**
>
> **A3**: Thanks for the reference! We discussed the limitations of current composition and manipulation methods in Section 2.3. (1) Both object-centric NeRF and Neural Scene Graphs require a shared MLP renderer for the objects to be composed, which means the objects trained in one scene cannot be directly composed with objects trained with another scene, due to the different MLP renderers. Differently, our method supports natural composition without such constraints in training. (2) We focus on rigid transformation and composition of different NeRF models, while Edit-NeRF mainly focus on editing a single NeRF model. Our composition requires no extra optimization, while Edit-NeRF still requires an optimization process to apply the manipulations. We think these two types of NeRF manipulation are complementary, and both contribute to making NeRF manipulatable as triangular meshes.
>
> **Q4: Ablation study and better analysis on the explicit control of model capacity by increasing rank components.**
>
> **A4**: Thanks for the advice! We analyzed the influence of different numbers of rank components to the model capacity in Figure 3. We found that the proposed rank-residual learning successfully controls the growth of model capacity by increasing the number of rank components, achieving near-optimal performance. We further perform an ablation study to analyze the best training strategy to exploit the increasing model capacity. Assume we have a set of rank groups for different model capacities, we experimented with three different settings: (1) sequentially training each rank group until its convergence, freezing the previous rank groups before training a new rank group, so each loss only applies on its corresponding rank group, (2) parallel training all rank groups, but for each group we detach the output from the previous groups, so each loss still applies independently, (3) parallel training all rank groups without detaching, so each loss applies to all the previous rank groups (the setting in our paper).
>
> | Settings                       | PSNR  | Training Time (min) |
> | ------------------------------ | :---: | :-----------------: |
> | (1) sequential                 | 34.16 |         83          |
> | (2) parallel w/ detach         | 33.69 |         28          |
> | (3) parallel w/o detach (ours) | 34.37 |         26          |
>
> The first setting requires longer training time to assure convergence of each rank group. We find the third setting achieves better performance and faster convergence compared to the first two settings. We think the parallel training without detaching encourages the later rank groups to learn complex details and the earlier rank groups to focus on the fundamentals.
>
>
>
> [1] TensoRF: Tensorial Radiance Fields. Chen et. al.

---

> > ### Comment · Reviewer_x5VX · 2022-08-08
> > **Responses**
> >
> > Thank the authors for the detailed responses.
> > Most of my concerns are resolved.
> >
> > It is ok to accept this paper, but rejecting it would not be too bad, considering that it shares similar ideas with TensoRF on the tensor decomposition and has worse performance.

---

> > > ### Author Response · Authors · 2022-08-08
> > > **Response to reviewer x5VX**
> > >
> > > Thanks for the response!
> > >
> > > As acknowledged by the reviewer, we propose a novel method to dynamically compress a NeRF and adjust the levels of details without extra optimization.
> > >
> > > Our model further explores the low-rank approximation property based on TensoRF's tensor rank decomposition. The extra compressibility and composability come at the price of slightly worse performance (PSNR -0.77, or -2.3%), which is hard to discern in visual comparisons.

---

> > > > ### Comment · Reviewer_x5VX · 2022-08-09
> > > > **Responses**
> > > >
> > > > Yes, I have considered the things pointed by you as the paper's technical contributions. So I do not vote to reject this paper.
> > > >
> > > > Actually, TensoRF can also dynamically compress a NeRF and adjust the levels of details, because it is also based on the tensor decomposition.
> > > >
> > > > It seems that the compressibility of the proposed approach is not better than that of TensoRF. The composability of this paper should be the same as TensoRF in theory.
> > > >
> > > > Please do not overclaim the advantages of the proposed method, which could harm this community.

---

> > > > > ### Author Response · Authors · 2022-08-09
> > > > > **Response to reviewer x5VX**
> > > > >
> > > > > Thanks for the response! We greatly appreciate your time and effort in the review and discussion.
> > > > >
> > > > > We are not trying to overclaim our advantages, but want to point out some misunderstandings in the last response:
> > > > >
> > > > > (1) **TensoRF requires a full retraining to change the model size, because their tensor rank decomposition does not hold the low-rank approximation property**. Our contribution is the rank-residual training strategy to ensure the low-rank approximation property for the learnt decomposition, such that **we can compress the trained model without extra optimization**.
> > > > >
> > > > > (2) For composability, our motivation is to remove the MLP renderers used in hybrid methods including TensoRF. To achieve composition of different models, **TensoRF requires a shared MLP renderer for these models, limiting the potential to extend to other models. By removing the MLPs, our method gets rid of such restrictions in training.**

---

> > > > > > ### Comment · Reviewer_x5VX · 2022-08-09
> > > > > > **Responses**
> > > > > >
> > > > > > 1. TensoRF hold the low-rank approximation property.
> > > > > > 2. TensoRF can be easily revised to support the controllability. Both CCNeRF and TensoRF are based on tensor decomposition. Their behaviors should be very similar.
> > > > > > 3. I acknowledge the rank-residual training strategy, but it is straightforward.
> > > > > > 4. TensoRF works well without a MLP renderer.
> > > > > >
> > > > > > The core reason that I currently feel positive about this paper is that TensoRF does not get accepted to ECCV before the NeurIPS submission  deadline.
> > > > > >
> > > > > > From my perspective, this paper adds some straightforward tricks to improve TensoRF:
> > > > > > 1. Change the CP decomposition to Tucker decomposition, which supports the controllability of the model size.
> > > > > > 2. Rank-residual training strategy.
> > > > > >
> > > > > > Both tricks are straightforward. More imporatnt, this paper has worse performance than TensoRF in terms of the model size and training time.

---

> > > > > > > ### Author Response · Authors · 2022-08-09
> > > > > > > **Response to Reviewer x5VX**
> > > > > > >
> > > > > > > **Q1**: TensoRF holds the low-rank approximation property.
> > > > > > >
> > > > > > > **A1**: By low-rank approximation, we refer to the optimal approximation of a high rank matrix with a low rank matrix (i.e., the Eckart–Young–Mirsky theorem of SVD), or at least a near-optimal one. General tensor rank decomposition (CP, Tucker) does not hold such property [1]. TensoRF learns a similar decomposition from data, and also does not hold the low-rank approximation property too. We discussed and demonstrated the performance degeneration when empirically truncating our baseline model that is similar to TensoRF in Figure 3. The major cause of such degeneration is the scattered distribution of rank importance (Figure 4). Therefore, we propose the rank-residual training to force the rank importance to concentrate on the lower ranks. **This rank-residual training is the crucial component to achieve near-optimal low-rank approximation.**
> > > > > > >
> > > > > > > **Q2**: This paper adds some straightforward tricks to improve TensoRF.
> > > > > > >
> > > > > > > **A2**: We consider that "straightforward" should not be considered as a disadvantage. Although these ideas seem straightforward, our work first adopted them to achieve dynamic compressibility and composability, which are important for practical applications of NeRF, but not well discussed in previous works.
> > > > > > >
> > > > > > > **Q3**: Change the CP decomposition to Tucker decomposition, which supports the controllability of the model size.
> > > > > > >
> > > > > > > **A3**: We are not changing to the Tucker decomposition, although the CP decomposition can be viewed as a special case of the Tucker decomposition. Instead, we use a mixture of the CP and Tri-plane (TP) decomposition.
> > > > > > >
> > > > > > > **Q4**: This paper has worse performance than TensoRF in terms of the model size and training time.
> > > > > > >
> > > > > > > **A4**: We emphasize again that our aim is not to achieve SOTA, but demonstrate how the extra compressibility and composability can help in practical applications of NeRF.
> > > > > > >
> > > > > > >
> > > > > > > [1] Tensor Decompositions and Applications, Kolda et. al.

---

### Official Review · Reviewer_i1vi · 2022-07-12

**Rating:** 6
**Confidence:** 4
**Soundness:** 3 good
**Presentation:** 3 good
**Contribution:** 3 good

**Summary:**

This paper proposes a rank-residual learning strategy to obtain a radiance field representation that supports compressibility and compositionality with an acceptable diminishment of rendering quality.


**Questions:**

- What’s the training and inference time of the proposed method?


**Limitations:**

No.

**Strengths And Weaknesses:**

### Strengths

*Originality*:
- I think this work is a nice combination of decomposition and Neural Radiance Fields (NeRFs).
- The capability column of Table 2 clearly illustrates the difference between this work and existing works, demonstrating the new capabilities enabled by the proposed method.
*Clarity*
- The submission is well-written and easy to follow.
- Related works are adequately cited and compared. For example, Line 103 discusses the similarities and differences between this work and TensoRF clearly.
*Quality*:
- The proposed approach is technically sound.
- The claims are well supported by the experimental results.
- The authors are careful and honest with the limitations (e.g., the model size and rendering time grow linearly with the total number of ranks, baked lighting, and bounded scene).

### Weaknesses

*Clarity*
- Line 70-90, I think categorizing methods into a) neural network-based, b) hybrid, and c) no neural network is more accurate than a) implicit, b) hybrid, and c) implicit. Some 3D representations are called **implicit** for years even if they are stored in voxels, e.g., TSDF, where the signed distance function is actually an implicit shape function.
- I think Figure 2 needs more annotations. For example, what’s the meaning of dotted lines in the middle sub-figure? What’s the meaning of black lines in the middle and right sub-figures? I assume the meanings of those lines are weighted interpolations. It would be nice to confirm that.

---

> ### Author Response · Authors · 2022-08-02
> **Reply to reviewer i1vi**
>
>  We thank the reviewer for the constructive suggestions. Below are our responses to the questions.
>
> **Q1: Categorization of NeRF methods.**
>
> **A1**: Thanks for the advice! We have renamed the categories in the revised version.
>
> **Q2: More annotations for Figure 2.**
>
> **A2**: Yes, the black lines mean we first project the 3D point to the decomposed line or plane, and then perform interpolations to calculate the features for the 3D point. The dotted lines are auxiliary lines to make the projection more clear. We have added more annotations in the revised version.
>
> **Q3: Training and Inference time of the proposed method.**
>
> **A3**: We compare the average training time of our method with recent methods on the NeRF-synthetic dataset (measured with a V100 GPU):
>
> | Methods       | Ours-CP | Ours-HY-S | Ours-HY | NeRF | TensoRF-CP-384 | TensoRF-VM-192 |
> | ------------- | :-----: | :-------: | :-----: | ---: | :------------: | :------------: |
> | Training time | 29 min  |  30 min   | 41 min  | 35 h |     25 min     |     17 min     |
>
> The inference speed of our method is highly dependent on the complexity of the scene. We measure the inference time to render an 800x800 image for a scene with different composed objects (as in the teaser image):
>
> | Settings       | Hotdog | Hotdog + Ficus | Hotdog + Ficus + 3 chairs |
> | -------------- | :----: | :------------: | :-----------------------: |
> | Inference time | 2.22 s |     3.27 s     |          6.62 s           |
>
> Although efficient training and inference is not the major topic of our method, we still achieve considerably faster training and inference speed compared to the vanilla NeRF, due to the no-neural-network model we use. Compared to TensoRF, our method trains slightly slower due to the extra computation of the rank-residual loss.

---

> > ### Comment · Reviewer_i1vi · 2022-08-09
> > **Responses**
> >
> > Thanks for the answer. The authors have addressed my concerns and I will keep my score the same.

---

### Official Review · Reviewer_46de · 2022-07-12

**Rating:** 4
**Confidence:** 4
**Soundness:** 3 good
**Presentation:** 3 good
**Contribution:** 2 fair

**Summary:**

This paper presents a compressible NeRF model which also supports the composition of different NeRFs to form a new scene. The proposed method uses a mixture of CANDECOMP/PARAFAC (CP) decomposition and Triple Plane (TP) decomposition, which are vector-based and matrix-based respectively, contributing to a hybrid decomposition method. At the same time, in order to ensure that the selected rank components can achieve near-optimal compression results, the authors propose a rank-residual learning strategy. The entire model does not use the MLP network, and different LODs can be achieved by selecting different numbers of rank components. The rank components of different scenes are concatenated together to achieve the composition of different NeRFs.

**Questions:**

-- As for composition, combining different NeRF models does not pose a particular technical challenge. One can obtain a rough geometric bounding box after the training is completed, and then combine the bounding boxes to render combined images. So in Table 2, why can’t some methods, such as PlenOctree, achieve composition?
--What is the difference between the composition that this method can achieve compared to other methods such as NSVF and Plenoxels?
--In Table 2, why only the proposed method can compress the NeRF model? Due to it can change the compression ratio? Similar method, TensoRF, can also achieve a small model size. What’s the advantage of the proposed method?
-- TensoRF uses the MLP network to solve the problem of worse performance caused by Sphere Harmonics (SH). The proposed method also uses the SH functions, but does not use MLP. How does the proposed method solve the problem of worse performance?


**Limitations:**

The authors have discussed the limitations.

**Strengths And Weaknesses:**

Strengths
--The decomposition method is novel, although in some aspects, such as CP decomposition, are the same as the existing work TensoRF. But this work is based on low-rank approximation, the proposed hybrid decomposition strategy can adjust the ratio of vector- and matrix-based rank components, which is something that TensoRF can't do.
--The method can adjust the model size, and level of detail.
--The paper is clear and easy to understand.

Weaknesses
-- Although in terms of method, the authors discuss the differences with TensoRF in the supplementary material, it is undeniable that the motivation of the two works is very similar (both introduce vector-based or matrix-based decomposition into NeRF). And in the comparison of TensoRF, the proposed method does not show superiority. In Table 2, the model size of method ‘TensoRF-CP-384’ is 3.9MB, which is smaller than ‘Ours-CP’ (4.4MB), while the quality is also better than ‘Ours-CP’. What’s more, the quality of ‘TensoRF-CP-384’ (3.9MB) is even slightly better than ‘Ours-HY-S’ whose model size is 68.9MB. Therefore, it is hard to illustrate the superiority of the proposed method in terms of both compression and rendering quality.

---

> ### Author Response · Authors · 2022-08-02
> **Reply to reviewer 46de**
>
> We thank the reviewer for the constructive suggestions. Below are our responses to the questions.
>
> **Q1: Why can only the proposed method compress the NeRF model?**
>
> **A1**: By 'compressible' we refer to the dynamic adjustment of model levels of detail without extra optimization, instead of the compactness of the model to represent a 3D scene. The main difference of our method from previous methods is that we can dynamically adjust the compression ratio on scenes by truncating the rank components, without retraining new models or additional fine-tuning. This property can be useful in many practical applications, such as the adaptive adjustment of model size to save memory without expensive retraining, and the progressive loading in network streaming.
>
> **Q2: Combining different NeRF models does not pose a particular technical challenge. Why can't PlenOctree achieve composition?**
>
> **A2**: Our motivation on composability is to achieve natural and easy composition of NeRF models like the widely used triangular meshes. Since implicit and hybrid NeRFs use MLPs to encode the scene, it is inconvenient and counterintuitive to record lots of MLPs to achieve composition. Instead, explicit NeRFs encode the scene in 3D volumes, which is natural for performing composition. Furthermore, our method proposes a new perspective on the composition of different models. The composition can be interpreted as the concatenation of different models' rank components. Besides, we can adjust the rank components for these models to achieve better rendering efficiency, especially for scenarios with multiple objects (eg. best illustrated in Figure 1 of the supplementary materials). PlenOctree contains two different stages, the NeRF-SH stage applies MLP but the octree stage only contains evaluated density and SH coefficients. The octree stage contains no neural networks and can be composed. We have corrected this in the revised version. Thanks for pointing this out!
>
> **Q3: What is the difference between the composition that this method can achieve compared to other methods such as NSVF and Plenoxels?**
>
> **A3**: (1) The main difference between the composition of our method and NSVF is that our method doesn't require a shared MLP renderer for different models. Methods like NSVF usually contain an MLP renderer to produce the density and color. As discussed in NSVF, different models (i.e., the sparse voxel volumes) must be trained with the same MLP renderer in order to be composed together. This limits the potential possibility to compose a wide range of models. (2) Although Plenoxels requires no MLP renderers, its model size is significantly larger (778MB on average). This large storage cost can hinder its practical usage when composing lots of models. Instead, our method can dynamically adjust the model size from 2.7MB to 88MB to control the total storage cost of a composed scene.
>
> **Q4: What’s the advantage of the proposed method over TensoRF?**
>
> **A4**: As clarified above, our motivation is to explore a naturally composable and compressible NeRF representation, instead of achieving SOTA performance in novel view synthesis. Compared to TensoRF and other NeRF models, our model can further (1) dynamically adjust the model levels of detail without extra optimization, (2) compose multiple single NeRF models into one scene without constraints in training. We believe these capabilities are important to make NeRF manipulatable as triangular meshes, and facilitate NeRF-based scene representation in practical applications.
>
> **Q5: How does the proposed method solve the problem of worse performance without MLP?**
>
> **A5**: (1) The major motivation for our paper is not to achieve SOTA performance in novel view synthesis. By removing MLP, we enjoy the natural composition of different models. (2) We are not the first to remove MLP and only use Spherical Harmonics (SH) to model the 3D scene. Plenoxels [1] also adopts a no-neural-network formulation.  (3) We would like to highlight that the worse performance is only relative. As shown in Table 2, our HY model's performance is better than many works that use MLPs (e.g., the vanilla NeRF, NSVF, and DVGO). (4) In general, our model can use more rank components to improve the performance as a remedy.
>
>
>
> [1] Plenoxels: Radiance fields without neural networks. Sara Fridovich-Keil and Alex Yu et. al.

---

> > ### Comment · Reviewer_46de · 2022-08-08
> > **Response to authors**
> >
> > I do agree that the proposed method can dynamically adjust the model levels of detail without extra optimization, while as far as I know, there is no other method that can do this. Also, the method can better support the composition of NeRF. However, I still argue the validity of the method. Although the goal of this method is not to achieve a better quality of novel view synthesis, there should be some guarantee of synthesis quality as the model size decreases. But as I pointed out before, the PSNR value of the proposed method is slightly lower than that of TensoRF at a larger model size. Although compared with TensoRF, the proposed method does not use MLP, the rebuttal shows that the training time is longer than that of TensoRF. It should also be noted that vanilla NeRF can achieve good synthesis quality with a model size of only 5MB, while the Ours-HY-S model is already 68.9MB.

---

> > > ### Author Response · Authors · 2022-08-08
> > > **Response to reviewer 46de**
> > >
> > > Thanks for the response! We would like to clarify the validity and practicality of our method as following:
> > >
> > > (1) The quality for novel view synthesis is well guaranteed when we decrease the model size. The proposed residual-rank learning reaches similar performance to the best-possible model at the same size (Figure 3). As pointed out by the reviewer, **our PSNR is only slightly lower (-0.77, or 2.3%) compared to TensoRF, which does not have our extra compressibility and composability**. **We demonstrate that this PSNR drop is already hard to discern in qualitative comparisons** (Figure 3 of the supplementary materials).
> > >
> > > (2) We claim that it's **unfair to simply compare the model size of our method with the vanilla NeRF**. The vanilla NeRF belongs to the implicit NeRFs, which have a small model size (5MB) but take a long time to train (36 hours). Instead, our method belongs to the explicit NeRFs which train much faster (20-40 minutes), at the cost of a larger model size (e.g., Plenoxels at 778MB, DVGO at 612MB, and TensoRF at 72MB). Furthermore, **our dynamic compressibility provides a novel way to alleviate the growth of model size , by allowing us to adjust the model size from a large range of without optimization (e.g., 2MB to 69MB for the HY-S model)**.
> > >
> > > In conclusion, the extra compressibility and composability of our method does come at a price of performance, but we carefully discussed and analyzed these limitations in our paper. **As acknowledged by all reviewers, we propose a novel perspective to explore the dynamic compressibility and compositionality of NeRF representations.** Therefore, we still consider our contributions to outweigh the limitations, and sincerely hope the reviewer can improve the rating.

---

### Official Review · Reviewer_rjyH · 2022-07-12

**Rating:** 6
**Confidence:** 2
**Soundness:** 2 fair
**Presentation:** 2 fair
**Contribution:** 2 fair

**Summary:**


This paper proposes an MLP-free NeRF representation that supports both compression and composability. The representation allows for efficient and convenient manipulation of the scene of interest and resembles the Level of Detail (LOD) concept in computer graphics. The ability to compose multiple scenes or objects comes naturally from this representation.

With no neural network involved, the model represents the radiance field of a scene using tensors that are the tensor rank decomposition of a full tensor explicitly expressing the radiance field. The authors study two types of tensor decomposition allowing the user to control model sizes. A rank-residual learning strategy is used to support an easy trade-off between model size and quality, without any re-training.

The authors use a hybrid feature volume decomposition that mixes the more expensive CP decomposition and the more compact TP decomposition. The mixing ratio controls model size vs. quality.

Compared with spatial locations XYZs, viewing directions are less “native” to the voxel grid (or tensor) representation, so the authors represent them using the Spherical Harmonics (SH) functions.

Because the CP decomposition does not come with the property of “rank importance,” the authors propose a rank-residual learning strategy to learn the coefficient residuals either sequentially or in parallel.

====== POST-REBUTTAL UPDATE ======

I read the authors' rebuttal that addresses my concerns reasonably. I appreciate the extra experiments, too. Overall, I'm willing to raise my rating to Weak Accept, to the best of my non-expert knowledge. This of course is conditioned on that the authors will add these new experiments to their final version and add necessary clarifications.

**Questions:**



I understand how the rank residual learning works for the sequential case but want the authors to clarify how the parallel computation works for this case. In other words, would a parallel computation lead to exactly the same set of coefficients as a sequential computation does?

Which of CP and TP decompositions does a better job of modeling specularities?

Are there special decomposition constraints that can be applied to the decomposition such that view-dependent effects are prioritized and preserved?

Also related to view-dependent effects, I think the higher degree you use for SH, the more accurate view-dependent effects you get since the view directions are more “concentrated” and hence more accurate. But with an SH degree of 4, I don’t think you will be able to get smoothly moving specularities. Have the authors tried higher degrees (I’m aware of the squared growth of the number of SH coefficients)? This sounds like a meaningful ablation study to do.




**Limitations:**

Yes, the authors mention baked-in lighting as a limitation.

**Strengths And Weaknesses:**


The paper offers a fresh perspective of NeRF, similar to Plenoxels’: radiance fields are not necessarily represented by a neural network. A full voxel grid representation is either low-resolution or expensive, but using tensor decomposition ameliorates this problem.

The major weakness of this paper is that it has not studied view-dependent effects such as specular highlights. As the figure and video of the “drums” scene show, the compressed model struggles to produce photorealistic or smoothly varying specularities. Since the main use of NeRF is view synthesis where view-dependent effects are the first-class citizen, studying how the tensor decomposition methods affect modeling view-dependent effects. For instance, which of CP and TP decompositions does a better job of modeling specularities? Are there special decomposition constraints that can be applied to the decomposition such that view-dependent effects are prioritized and preserved? Given the authors consider just NeRF as the application, I don’t think this can be neglected by this paper.

---

> ### Author Response · Authors · 2022-08-02
> **Reply to reviewer rjyH**
>
> We thank the reviewer for the constructive suggestions. Below are our responses to the questions.
>
> **Q1: Parallel computation's influence on the learned coefficients.**
>
> **A1**: The parallel training cannot lead to exactly the same coefficients as the sequential training. In fact, we find that the parallel training has slightly better performance. We experimented with three settings on the chair dataset: (1) sequentially training per stage until its convergence, freezing the previous stages before training a new stage, (2) parallel training all stages, but for each stage we detach the output from the previous stages so each loss only applies on its corresponding rank group, which can be viewed as training each stage independently in parallel, (3) parallel training all stages without detaching, so each loss applies to all its previous rank groups (the setting in our paper):
>
> | Settings                       | PSNR  | Training Time (min) |
> | ------------------------------ | :---: | :-----------------: |
> | (1) sequential                 | 34.16 |         83          |
> | (2) parallel w/ detach         | 33.69 |         28          |
> | (3) parallel w/o detach (ours) | 34.37 |         26          |
>
> The first setting is significantly slower to assure convergence of each stage. The second setting's final performance is worse due to optimizing later stages with not fully converged earlier stages. Compared to the first two settings, we think the third setting eases the training of the earlier stages, by letting the later stages model the complex details. Therefore, the earlier stages can focus on the fundamentals.
>
> **Q2: The compressed model struggles to produce photorealistic or smoothly varying specularities.**
>
> **A2**: We would like to highlight that our model with all rank components is capable of producing photorealistic renderings. We are not the first to adopt only Spherical Harmonics (SH) without MLP to model the specularities. Plenoxels [1] only use 9 terms SH without MLP to model complex scenes. As compared in Table 2, our HY model achieves better performance and takes much less storage compared to Plenoxels. For the compressed models, since we perform lossy low-rank approximation by truncating the full model, the specularities do get harmed. However, we still consider it is more favorable compared to harming the diffuse color, as shown in the baseline model.
>
> **Q3: Which of CP and TP decompositions does a better job of modeling specularities?**
>
> **A3**: With enough rank components, both CP and TP decomposition can model the specularities well. In general, the TP method contains more parameters per rank and the model capability is better. Therefore, it requires fewer rank components to model the same level of specularities.
>
> **Q4: Are there special decomposition constraints that can be applied to the decomposition such that view-dependent effects are prioritized and preserved?**
>
> **A4**: This is an interesting idea. We have experimented on some decomposition constraints such as weight normalization and orthogonal regularization, but it is unclear how these general decomposition constraints can be connected to prioritizing view-dependent effects, which is quite task-specific. We consider it as a future direction to explore.
>
> **Q5: Ablation study on the influence of SH degrees to specularities.**
>
> **A5**: Thanks for the advice! We performed an ablation study on the SH degrees with the materials dataset, which contains lots of view-dependent effects:
>
> | SH terms            |   4   |   9   | 16 (ours) |  25   |
> | ------------------- | :---: | :---: | :-------: | :---: |
> | PSNR                | 28.29 | 28.97 |   28.99   | 28.65 |
> | Training Time (min) |  54   |  44   |    49     |  73   |
>
> We show that 16 terms SH achieves the best PSNR to model view-dependent effects. With too few or too many terms (e.g., 4 and 25), the model is hard to converge and takes more time to train. Plenoxels [1] only uses 9 terms SH. We found 16 terms SH could slightly improve the performance without making the convergence significantly slower.
>
>
>
> [1] Plenoxels: Radiance fields without neural networks. Sara Fridovich-Keil and Alex Yu et. al.

---

### Author Response · Authors · 2022-08-08
**A reminder for the discussion**

Dear AC and all reviewers:

Thanks again for all of your constructive suggestions, which have helped us improve the quality and clarity of the paper!

Since the discussion phase has only one day left and we have not heard any post-rebuttal response yet, please don’t hesitate to let us know if there are any additional clarifications or experiments that we can offer, as we would love to convince you of the merits of the paper. We appreciate your suggestions. Thanks!

---

### Comment · Area_Chair_dbrZ · 2022-08-09
**reminder for discussion**

Dear reviewers,

Thank you all for providing valuable comments. The authors have provided detailed responses to your comments. Has the response addressed your concerns?

If you haven't, I would appreciate it a lot if you could reply to the authors’ responses soon as the deadline is approaching (Tues, Aug 9).

Best,

ACs

---

### Meta-Review · Area_Chair_dbrZ · 2022-08-29

**Recommendation:** Accept
**Confidence:** Certain

**Metareview:**

This paper presents a new NeRF method based on tensor decomposition. The method supports both compression and composability, while achieving similar results compared to standard NeRF models. The method does not use a neural network. Several reviewers found the paper easy to follow, the method novel & sound, and the comparisons comprehensive. Two reviewers mentioned the similarity between the proposed work and TensoRF. The rebuttal addressed most concerns and highlighted the differences between the two works. As TensoRF is a concurrent ECCV submission, the existence of TensoRF should not be used against the proposed work. The AC agreed with most of the reviewers and recommended accepting the paper.


**Award:**

No

---

### Decision · Program_Chairs · 2022-09-14

Accept